# Global Flood Exposure from Different Sized Rivers

Mark V Bernhofen[1], Mark A Trigg[1], P Andrew Sleigh[1], Christopher C Sampson[2], Andrew M Smith[2]

[1]School of Civil Engineering, University of Leeds, LS2 9JT, United Kingdom
[2]Fathom, Square Works, 17-18 Berkeley Square, BS8 1HB, United Kingdom

*Correspondence to*: Mark V Bernhofen (cn13mvb@leeds.ac.uk)

**Abstract.** There is now a wealth of data to calculate global flood exposure. Available datasets differ in detail and representation of both global population distribution and global flood hazard. Previous studies of global flood risk have used datasets interchangeably without addressing the impacts using different datasets could have on exposure estimates. By calculating flood exposure to different sized rivers using a model independent geomorphological River Flood Susceptibility

Map (RFSM), we show that limits placed on the size of river represented in global flood models result in global flood exposure estimates that differ by greater than a factor of 2. The choice of population dataset is found to be equally important and can have enormous impacts on national flood exposure estimates. Up-to-date, high resolution population data is vital for accurately representing exposure to smaller rivers and will be key in improving the global flood risk picture. Our results inform the appropriate application of these datasets and where further development and research is needed.

## 1 Introduction

River floods are amongst the most frequent and damaging natural disasters globally (Wallemacq et al., 2015). Considerable effort has gone into understanding global river flooding over the last decade, and a number of global flood models (GFMs) have been developed concurrently (Yamazaki et al., 2011, Pappenberger et al., 2012, Winsemius et al., 2013, Rudari et al., 2015, Sampson et al., 2015, Dottori et al., 2016c). The usefulness of these GFMs was initially limited to

coarse scale flood risk assessments (Ward et al., 2015), largely due to global-scale data limitations. However, the incorporation of higher accuracy terrain data, available at the national level, has shown that their modelling frameworks are also suited to identifying more localized risk when utilising local data (Wing et al., 2017). Previous studies comparing GFMs have shown there is disagreement between the global flood extents (Trigg et al., 2016b, Bernhofen et al., 2018b, Aerts et al., 2020). This disagreement between GFMs stems from different model structures and methods. One key difference between

the models, which has not yet been explored, is the size of their river networks. The models have different river size thresholds at which they simulate fluvial events. These thresholds determine the size, and number, of rivers represented in GFMs, which can differ by several orders of magnitude. The size of a model's river network is contingent on both the quality and resolution of the model input datasets such as the underlying digital elevation model (DEM) and climatology (Dottori et al., 2016c) as well as the computational efficiency of the model, as the introduction of smaller rivers

exponentially increases the modelled domain. Chosen thresholds also influence estimates of global flood exposure, as larger river networks result in higher simulated flood volumes and potential exposure. The effect that GFM river network size has

on flood exposure estimates has not yet been quantified at the global scale. As Remote Sensing (RS) technologies continue to advance, so will the granularity at which rivers can be represented globally. Smaller rivers, previously unrepresented in coarse global datasets, will be able to be studied and modelled at large scales; potentially reframing current global flood exposure estimates. Limited work has been dedicated to the investigation of the human interaction with rivers of different size (Kummu et al., 2011). Understanding this interaction globally, particularly with respect to river flooding, will inform us about the completeness of current global flood exposure studies and identify where further study and development is needed.

A comprehensive understanding of flood risk requires information about the hazard, what or who is exposed, and their vulnerability. Exposure could include damages (both direct and indirect), exposed gross domestic product (GDP), exposed assets, and most commonly: exposed people (Ward et al., 2020). Identifying flood exposed populations usually involves intersecting a flood hazard map with a population map. The methods and inputs used to produce population datasets differ, and so does their intended use (Leyk et al., 2019). Recently released population maps, which utilize commercial RS data and are an order of magnitude more resolved than existing population datasets (Tiecke, 2017) are already being used for disaster preparedness and response (Facebook, 2019). However, our current understanding of global flood exposure is based on existing global population datasets, and these datasets have been used interchangeably in global studies (Tanoue et al., 2016, Jongman et al., 2012, Dottori et al., 2018) with little comment about their relative merit. The credibility of existing global flood exposure estimates in light of new, more detailed, population data and the implications of their interchangeable use in studies of global flood exposure needs to be explored. A recent study by Smith et al. (2019) reported large disagreement between flood exposure estimates calculated in 18 developing countries using three different population datasets. The identification of population data as one of the chief sources of uncertainty in global flood exposure studies warrants further investigation at the global scale. Understanding how both new and existing population datasets differ in their resulting exposure estimates, both regionally and within the hierarchy of the river network, can inform users about the most appropriate population dataset to use.

To explicitly explore the impact of river network size on global flood exposure estimates, we use a geomorphological measure of a river's flood susceptibility, which is independent from current GFMs and the additional uncertainties their different model structures bring. Fluvial processes contribute to the evolution of a landscape over time. The erosional action of flowing water has shaped the terrain of drainage basins to reflect the historical flow of water through them. Geomorphological approaches to mapping river flood susceptibility rely on the concept that the cumulative hydrogeomorphic effect of past flood events, evident in topography data, is indicative of a river's propensity to flood. Such approaches to flood mapping have been applied over a number of scales: from local (Nardi et al., 2006, Nobre et al., 2016, Dodov and Foufoula-Georgiou, 2006), to national (Jafarzadegan et al., 2018, Samela et al., 2017), to regional (Lugeri et al., 2010) and global (Nardi et al., 2019). The computational efficiency of geomorphic flood mapping, coupled with its reliance on only terrain data as input, make it useful for a 'first look' global scale analysis; intended to inform future development of higher accuracy hydrological flood mapping (Di Baldassarre et al., 2020).

Our geomorphological approach to mapping a river's flood susceptibility, herein referred to as the River Flood Susceptibility Map (RFSM) is based on new topography data (Yamazaki et al., 2017), which incorporates crowdsourced information to better represent the locations of rivers and streams (Yamazaki et al., 2019). Validation of our calibrated methodology (outlined in detail in the Supplementary Material) shows that the RFSM better replicates GFM hazard maps in Africa than an existing global geomorphological approach (Nardi et al., 2019). We also show that the RFSM performs

similarly to the best GFMs (Dottori et al., 2016c, Sampson et al., 2015, Yamazaki et al., 2011) when validated against historical flood events (Bernhofen et al., 2018b).  The RFSM allows us to easily discretize the flood map into different river sizes (independently of GFMs). We investigate the human interface with these different size rivers using three population datasets. Facebook's High Resolution Settlement Layer (HRSL),  (https://data.humdata.org/organization/facebook?q=density) (1 arc-second, ~30 m resolution at the equator) (Tiecke, 2017) which is currently only available in 168 countries globally,

and two population datasets used extensively in previous studies of global flood risk: the Global Human Settlement Population (GHS-POP) (http://doi.org/10.2905/0C6B9751-A71F-4062-830B-43C9F432370F) (9 arc-second, ~250 m resolution at the equator) (Freire et al., 2015) and WorldPop (https://dx.doi.org/10.5258/SOTON/WP00645) (3 arc-second, ~90 m resolution at the equator) (Stevens et al., 2015, Lloyd et al., 2019). We present a global picture of flood exposure to different size rivers, both in the present day, and how it has changed over the past 40 years. We then compare the flood

exposure calculated using different population layers, exploring the implications this has on national level flood exposure estimates and examine the impact that river size has on any disagreement. Finally, we address the size of rivers represented in GFMs specifically and investigate how their chosen river network size impacts both global and national flood exposure estimates and what implications this has for previously published global flood risk assessments.

## 2 Methods

### 2.1 Mapping River Flood Susceptibility

       We use a geomorphological approach to mapping river flood susceptibility, which is independent from the global flood models (GFMs). Previous GFM comparison studies found that multiple aspects of model structure contributed towards disagreement (Trigg et al., 2016b, Bernhofen et al., 2018b, Aerts et al., 2020). Using a geomorphological approach, we are able to explore just one aspect of disagreement: river network size. This approach allows us to explore all stream scales as

drainage paths can be identified from the terrain alone. It is not influenced by the structure of the different GFMs and does not have the same computational restraints as a global hydrodynamic model. This approach is different from the GFMs in that it does not measure the flood extent for a given return period flood, but rather a river and surrounding location's static susceptibility to flooding.

       There are different approaches to geomorphic floodplain mapping. Three approaches were compared on the Tiber

River in Central Italy by Manfreda et al. (2014). That study found that approaches utilizing morphological descriptors to delineate floodplains better replicate reference flood extents. The best morphological descriptor was found to be the relative

elevation difference to the nearest channel (H). In a follow up study, Samela et al. (2017) investigated eleven different morphological descriptors in the Ohio River basin and then tested the best performing descriptors across the conterminous United States. While H was amongst the best four descriptors, it was shown to be highly variable across basins. The study found that the best morphological descriptor was a geomorphic index which relates H to a function of the nearest channel's contributing area. The method we use for delineating a river's flood susceptibility is based on the Height Above Nearest Drainage (HAND) methodology developed by Nobre et al. (2011). We use a variable H value ($H_n$), which changes depending on the Strahler stream order (Strahler, 1957) of the flooded channel (where n is the Strahler stream order). This geomorphic approach, requiring only terrain data as input, is computationally efficient, and can be easily modified to produce auxiliary data layers.

Our method, referred to as the River Flood Susceptibility Map (RFSM) (Bernhofen et al., 2021), is illustrated in *Figure 1* and takes three gridded datasets as input: a digital elevation model (DEM), its derived drainage directions, and its upstream drainage area (UDA). We use MERIT hydro data (Yamazaki et al., 2019), a hydrography dataset based on the error improved SRTM DEM: MERIT DEM (Yamazaki et al., 2017). MERIT Hydro is an improvement on previously available global hydrography datasets such as HYDROSHEDs (Lehner et al., 2008) in terms of both spatial coverage and its representation of small streams. Its improved representation of small streams is enabled by its incorporation of global water body data and crowdsourced Open Street Map river data. This makes it particularly suited to this study, where we are interested in examining the flood susceptibility of rivers down to the smallest streams.

The river network is extracted from the upstream drainage area dataset by specifying a minimum threshold river size (in units of UDA). Identifying the headwater of a river is no trivial task, with regional and climatic factors playing a part (Montgomery and Dietrich, 1988, Tarboton et al., 1991). Previous work exploring optimal initiation thresholds for geomorphological floodplain mapping found that DEMs with a resolution of 1 arc second (~30 m) could use initiation thresholds less than 10 km² UDA. In the same study, a 3 arc second (~90 m) resolution DEM was used with a 100 km² UDA threshold (Annis et al., 2019). The MERIT Hydro data we use in this study has a resolution of 3 arc seconds (~90 m). But its incorporation of crowdsourced river data has optimized its representation of small streams and rivers. As such, we use a globally consistent river initiation threshold of 10 km² UDA for the RFSM. This is a large assumption, as in some locations globally there will be no visible channel at this location. However, we argue that removing areas of potential exposure to avoid overprediction in some areas goes against the premise of this study, which is to explore and identify 'missed' areas of exposure. The exposure calculations for small streams should therefore be interpreted with these limitations in mind.

Once the river network has been extracted, the rivers in the network are classified based on their Strahler stream orders (Strahler, 1957). The Strahler stream order is a dimensionless indicator of the magnitude of the river based on its hierarchy within the drainage basin.

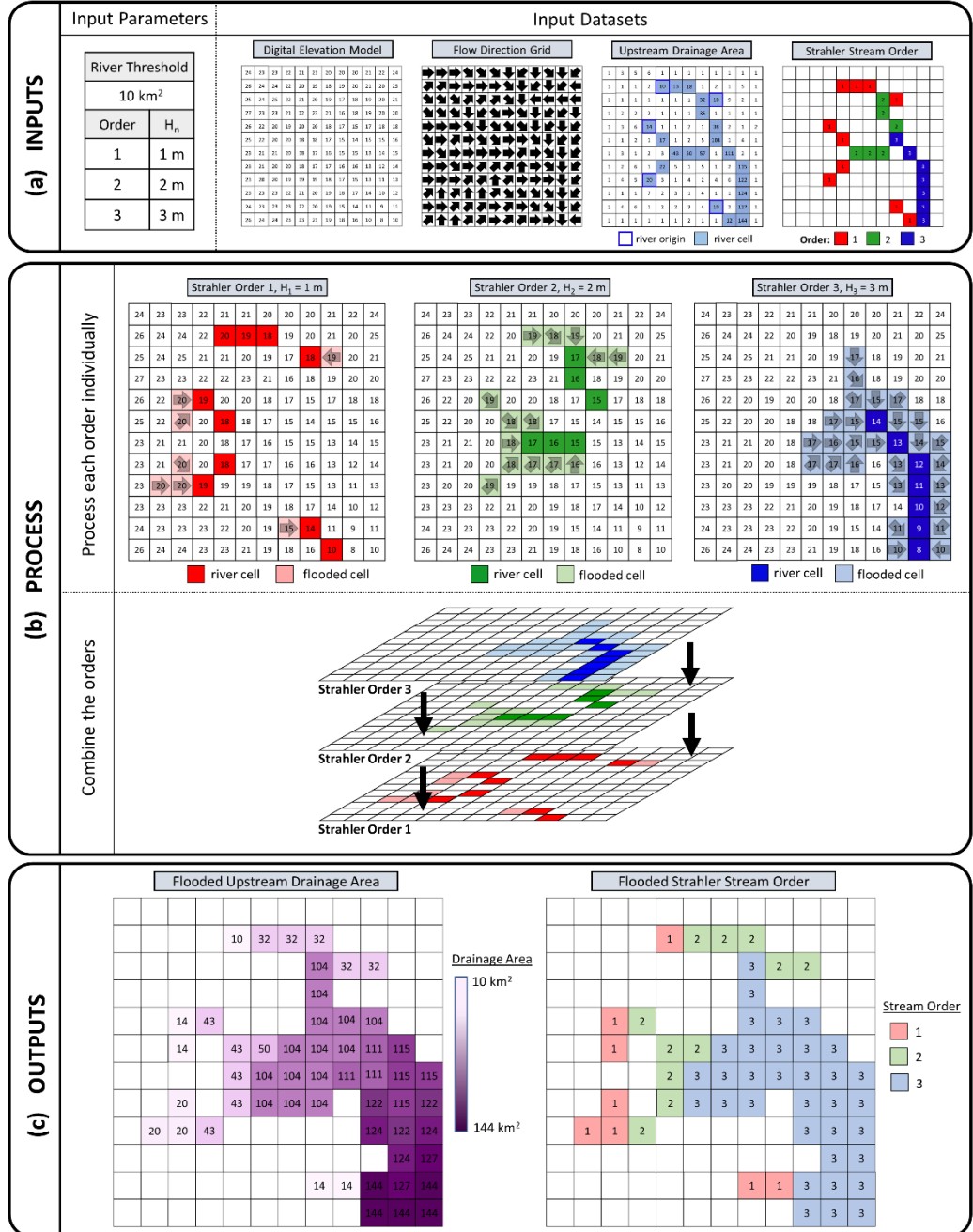

**Figure 1.** Illustrative example of the method for deriving the River Flood Susceptibility Map (RFSM). (a) User defined input parameters include the minimum river size and the maximum relative elevation difference to the nearest draining channel, $H_n$, for each Strahler stream order. Dataset inputs include a digital elevation model (DEM), flow direction grid, and an upstream drainage area grid (represented on a 12x12 1 km$^2$ grid for illustrative purposes). Rivers (as defined by the minimum river size threshold) are classified into Strahler stream orders. (b) Each Strahler stream order is processed separately using the Height Above Nearest Drainage (HAND) method and then the layers are combined. In areas of overlap the values for the highest order streams are retained. (c) Two outputs are produced: a map of the drainage area of the nearest flooded river and a map of the Strahler order of the nearest flooded river. See *Figure 7* for an example of RFSM outputs in Bosnia and Herzegovina and Guinea-Bissau.

### 2.1.1 Calibrating the River Flood Susceptibility Map

The maximum relative elevation difference to the nearest draining channel, $H_n$ (see *Figure 1a*), for each Strahler stream order (n) is the only RFSM parameter requiring calibration. We use a variable H, which scales with Strahler stream order, to account for changes in flood depth as a river's size changes. In Samela et al. (2017), the best performing geomorphic index also accounts for variations in river size by scaling relative to the river's upstream contributing area.

To account for climatic variability in a river's flood susceptibility (Smith et al., 2015), we split the globe into five simplified Köppen-Geiger climate zones (*Figure 2*): Tropical, Arid, Temperate, Continental and Polar. Polar regions are excluded from our analysis as these regions are dominated by glacial not fluvial processes (Chen et al., 2019). The RFSM has uniquely calibrated $H_n$ values in each of the four climate zones. We calibrate the $H_n$ values in 19 different basins (see *Figure 2*), spanning 5 different continents across all four climate zones considered. Reference flood maps used for calibration are a mixture of national, continental, and global flood hazard maps. To maintain consistency across the calibration data, we use 100-year return period flood hazard maps. We use a combination of national, continental, and global flood hazard maps for calibration in each climate zone. This is to ensure that there is sufficient calibration data for each Strahler order river, as only the national flood hazard data captures flooding for low order rivers. Two different national flood maps are used for calibration. The first is the National Flood Hazard Layer (NFHL) produced by the Federal Emergency Management Agency (FEMA) (https://www.fema.gov/flood-maps/tools-resources/flood-map-products/national-flood-hazard-layer). NFHL data is used for calibration in North American basins including Puerto Rico, Lower Gila, Upper Pecos, Lower Mississippi, Alabama, Muskingum, Rock, and Susquehanna. The second national flood map is the Environment Agency's 100-year flood map for planning (http://apps.environment-agency.gov.uk/wiyby/cy/151263.aspx), which is used for calibrating the RFSM in the Thames basin in England. The continental flood map for Europe (Dottori et al., 2016b), developed by the Joint Research Centre (JRC) is used to calibrate the RFSM in the Jucar river basin in Spain; the Loire river basin in France; the Po river basin in Italy and Switzerland; and the Oder river basin in Poland, Germany, and Czech Republic. A global flood hazard map (Dottori et al., 2016a), also developed by the JRC, is used to calibrate the RFSM in the Central Amazon basin in Brazil; the Lower Congo basin in the Democratic Republic of Congo and the Republic of Congo; the Lower Mekong basin in Thailand, Cambodia, Vietnam, and Laos; the Upper Nile basin in Egypt and Sudan, the Lower Lena basin in Russia and Kazakhstan; and the Central Lena basin in Russia. Maps of the reference flood maps used for calibration are shown in *Figure S1* and further details about each calibration basin can be found in *Table S1*.

The values are calibrated in each climate zone by running thousands of different combinations of $H_n$ in each calibration basin. Optimal $H_n$ values are determined by using three commonly used measure of fit scores: Critical Success Index (CSI), Hit Rate (HR), and Bias (Wilks, 2006). The $H_n$ values retained are the ones that result in the best fit scores with respect to the reference flood maps within each climate zone. Final calibrated $H_n$ values for each climate zone are shown in *Figure 2*. More detailed information on the calibration of the RFSM can be found in section S1 of the Supplementary Material.

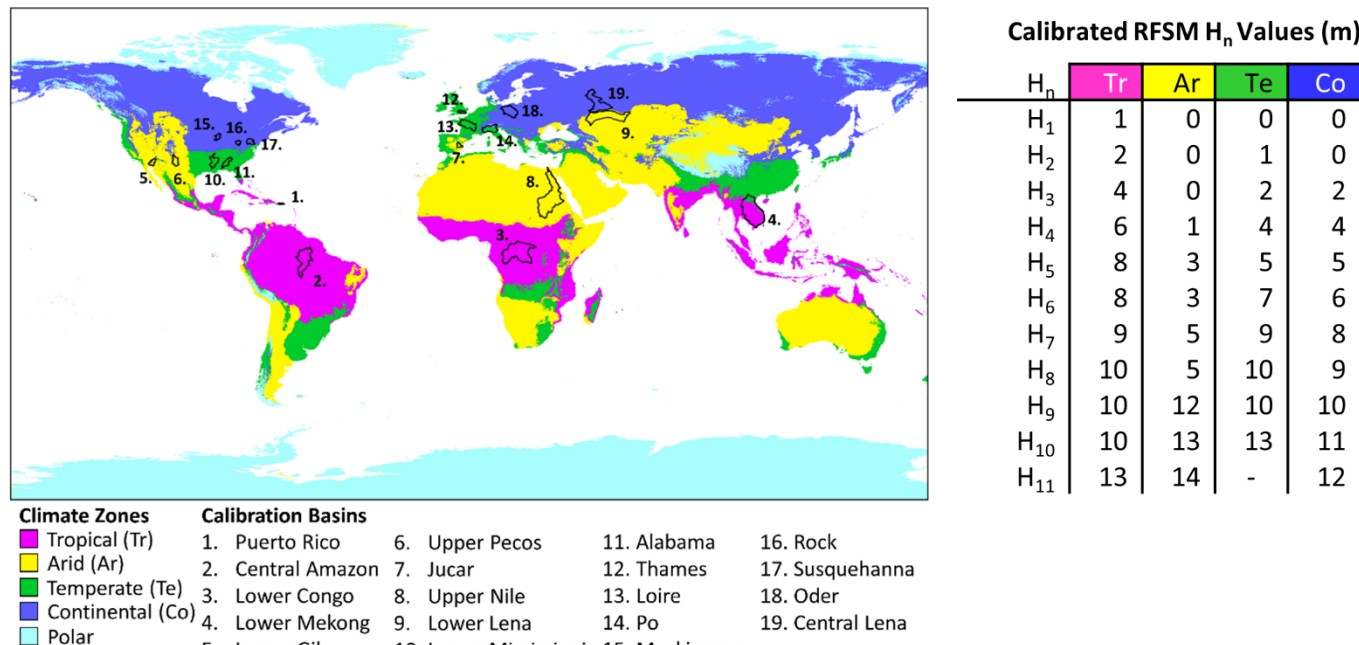

| $H_n$ | Tr | Ar | Te | Co |
|---|---|---|---|---|
| $H_1$ | 1 | 0 | 0 | 0 |
| $H_2$ | 2 | 0 | 1 | 0 |
| $H_3$ | 4 | 0 | 2 | 2 |
| $H_4$ | 6 | 1 | 4 | 4 |
| $H_5$ | 8 | 3 | 5 | 5 |
| $H_6$ | 8 | 3 | 7 | 6 |
| $H_7$ | 9 | 5 | 9 | 8 |
| $H_8$ | 10 | 5 | 10 | 9 |
| $H_9$ | 10 | 12 | 10 | 10 |
| $H_{10}$ | 10 | 13 | 13 | 11 |
| $H_{11}$ | 13 | 14 | - | 12 |

**Calibrated RFSM $H_n$ Values (m)**

**Climate Zones**
- Tropical (Tr)
- Arid (Ar)
- Temperate (Te)
- Continental (Co)
- Polar

**Calibration Basins**
1. Puerto Rico
2. Central Amazon
3. Lower Congo
4. Lower Mekong
5. Lower Gila
6. Upper Pecos
7. Jucar
8. Upper Nile
9. Lower Lena
10. Lower Mississippi
11. Alabama
12. Thames
13. Loire
14. Po
15. Muskingum
16. Rock
17. Susquehanna
18. Oder
19. Central Lena

**Figure 2.** The calibration basins shown on a map of the simplified Köppen Geiger climate zones and the calibrated maximum relative elevation difference to the nearest draining channel ($H_n$) for each Strahler stream order in the four climate zones considered (Polar regions are excluded from the analysis).

Once $H_n$ values for each order have been assigned, each stream order is processed separately (*Figure 1b)*, and then merged together. In areas of overlap, the highest order stream retains the values. Two datasets are produced as output: a map of the flooded river's upstream drainage area, and a map of the flooded river's Strahler stream order. Illustrations of these two outputs are shown in *Figure 1c*.

### 2.1.2 Validating the River Flood Susceptibility Map

The RFSM is validated against both existing GFMs and observed flood events. Validation against GFMs is carried out for the whole African continent using the 100-year return period aggregated output of six GFMs from a previous model intercomparison study (Trigg et al., 2016a). The six GFMs that make up the aggregated output include CIMA-UNEP (Rudari et al., 2015), Fathom (Sampson et al., 2015), GLOFRIS (Winsemius et al., 2013, Ward et al., 2013), JRC (Dottori et al., 2016c), and U-Tokyo (Yamazaki et al., 2011). To assess the credibility of the RFSM, it is also validated alongside an

existing global geomorphological floodplain map (Nardi et al., 2019). For validation we split the African continent into eight major drainage basins (see *Figure S3*) according to the HydroBasin Level 2 classification (Lehner and Grill, 2013). The results of the GFM validation show that the RFSM produces credible flood extents when compared with existing GFM outputs in Africa. The RFSM correctly captures over 90% of high agreement flood zones (where at least 5 out of 6 GFMs

agree) in 7 of the 8 major drainage basins in Africa. In the East African basin, the RFSM captures 87% of this high agreement flood zone. Comparing CSI, HR and Bias scores for the RFSM and the existing global geomorphological floodplain map, the RFSM scores better in all the major drainage basins in Africa except for North Africa (where both maps score poorly due to the Sahara Desert). The RFSM is also validated against observed flood events in Nigeria and Mozambique. The 2012 flooding in Nigeria and the 2007 floods in Mozambique affected four million people and over one hundred thousand people respectively (Bernhofen et al., 2018b). Validation data for both these flood events used in a previous GFM validation comparison study (Bernhofen et al., 2018a) is also used to validate the RFSM. The RFSM is validated against observed data in three validation regions: Lokoja, which is a narrow, confined floodplain at the confluence of Niger and Benue rivers in Nigeria; Idah, which is a flat and extensive floodplain south of Lokoja; and Chemba, which is an anabranching stretch of the Zambezi river just upstream of the delta in Mozambique. Validation of the RFSM against observed data from these historical flood events show that it performs similarly to the best performing GFMs in each of the three validation regions. Further detail about the validation of the RFSM can be found in section S2 of the Supplementary Material.

It is important to note the limitations of our methodology and geomorphological approaches in general. The RFSM does not account for flood protection measures and cannot communicate the probability of flooding in any location. It consistently represents a river's flood susceptibility based on the surrounding terrain alone. In regions where the floodplain boundaries are less distinguishable from the terrain, such as flat and low-lying areas, geomorphological approaches are prone to overprediction as they do not represent mass and momentum conservation. Our method's intended use is as a model-independent global 'first look' analysis to inform future hydrodynamic model development and use.

## 2.2 Measuring Exposure

We investigate the human exposure to river flood susceptibility. Human exposure is herein defined as the intersection of our river flood susceptibility map and a spatially distributed population layer. Three population datasets are used to measure exposure: Facebook's High Resolution Settlement Layer (HRSL) (https://data.humdata.org/organization/facebook?q=density) ) (Facebook and CIESIN, 2016), The European Commission Joint Research Centre's Global Human Settlement Population (GHS-POP) (https://dx.doi.org/10.2905/0C6B9751-A71F-4062-830B-43C9F432370F) (Schiavina, 2019), and WorldPop (https://dx.doi.org/10.5258/SOTON/WP00645) (Stevens et al., 2015). These population datasets all use the same initial input census data, from GPWv4 (Center for International Earth Science Information Network - CIESIN - Columbia University, 2016), but their methods for allocating the population across gridded cells differ. Facebook's HRSL is the only dataset of the three lacking full global coverage (at the time of writing 168 countries have been mapped). It is also the most recent, with work ongoing to map the remaining countries. HRSL uses ultra-high resolution commercial satellite imagery (~50 cm resolution) and convolutional neural networks to detect individual buildings at the country level (Tiecke, 2017). Subnational census data for the year 2018 is then proportionally allocated to the identified buildings at 1 arc second resolution (~30 m at the equator).

Similarly to the HRSL in methodology, JRC's GHS-POP dataset identifies built up areas from Landsat imagery and proportionally allocates census data to the built up areas (Freire et al., 2015). In regions where no settlements can be identified, but where census data indicates there is a population, the population is evenly distributed across the census area using areal weighting (Freire et al., 2016). This can occur in some rural areas, where small settlements are not captured by the Landsat imagery. Despite being coarser in spatial resolution at 9 arc seconds (~250 m at the equator), GHS-POP provides consistent multi-temporal population estimates (1975-1990-2000-2015) allowing for accurate analyses over time (Freire et al., 2020).

Unlike the other two population datasets, which evenly spread census data over identified settlements, WorldPop uses a complex model to disaggregate population over an area (Leyk et al., 2019). It uses a random forest model and a number of ancillary datasets to dynamically weight the distribution of census data over a 3 arc second (~90 m at the equator) gridded area (Stevens et al., 2015) to produce annual population estimates from 2000-2020.

Exposure calculations necessitate uniformity between the intersecting datasets in terms of spatial resolution. As such, the GHS-POP layer was resampled from 9 arc second resolution and the population evenly distributed to a 3 arc second resolution grid to allow for analysis with a flood map of the same resolution. Conversely, for the HRSL exposure calculations the RFSM was resampled from 3 arc second to 1 arc second resolution. When comparing the exposure results between population datasets the epoch used for comparison was 2015. National population totals for the HRSL and WorldPop datasets for the years 2018 and 2015, respectively, were scaled relative to GHS-POP 2015 national population totals.

## 3 Results and Discussion

### 3.1 Global Exposure to Different Sized Rivers from GHS-POP

Rivers were classified into 6 different sizes, expressed in upstream drainage area (UDA) ($km^2$), with the ranges increasing in powers of 10. River classifications based on UDA, depicted in *Figure 3b* for Nigeria, were as follows: stream (10-100 $km^2$), small river (100-1,000 $km^2$), medium river (1,000-10,000 $km^2$), medium-large river (10,000-100,000 $km^2$), large river (100,000-1,000,000 $km^2$), and huge river (>1,000,000 $km^2$).

Flood exposure is first calculated using the GHS-POP layer. Globally, we find 1.94 billion people susceptible to flooding from rivers with a UDA greater than 10 $km^2$. Breaking this down by continent, Asia's flood exposure is 1.49 billion, Africa's is 203 million, Europe's is 104 million, North America's is 81 million, South America's is 59 million, and Oceania's is 3.5 million. Splitting global flood exposure by river size, of the total exposed: 18.2% are from streams, 26.4% from small rivers, 23.7% from medium rivers, 17.2% from medium-large rivers, 8.4% from large rivers, and 6.1% from huge rivers. Asia makes up over 75% of the total global flood exposure, the majority of this amount coming from India and China, which are by far the two most exposed countries (see *Figure 3a*). Roughly half of India's flood exposure is from streams and small rivers. Comparably, in China, this figure is closer to a third. This is likely due to the degree of urbanisation in both countries; the percentage of China's urban population is double that of India's (WorldBank, 2018). Urban areas are disproportionately located on large rivers due to the historical tendency for settlements to form in areas fertile for farming and convenient for transport (McCool et al., 2009). As such, a greater proportion of flood exposure in China comes from larger rivers, whereas in India, a greater proportion comes from rural exposure to smaller rivers. Rivers classified as 'huge' are only found in some countries, but often they pose a large proportion of the national flood risk. For example, the Brahmaputra in Bangladesh and the Nile in Egypt and Sudan are responsible for just under half of the national flood exposure in their respective countries.

To identify countries with the most acute flood risk, exposure was normalized against total national population (*Figure 3c*). Suriname has the highest normalized exposure, with 894 people exposed per 1000. The country's low elevation relief, and its capital city situated on the banks of the Suriname river near its outlet into the Atlantic Ocean, makes Suriname particularly vulnerable to flooding (WorldBank, 2019). Four of the top 10 most 'normally' exposed countries are in south or south east Asia. These include Bangladesh, Cambodia, Thailand, and Vietnam. Flooding in these countries is severe and annual, normally occurring each year during the monsoon season. In Europe, the Netherlands has a high normalized exposure, 738 exposed per 1000. The Netherlands has a long history of flooding due to its low elevation, flat terrain, and high population density. It also has the most advanced flood defence systems in the world, designed to contain river water levels with a probability of occurrence once every 1250 years (Stokkom et al., 2005). Geomorphological approaches to flood mapping, such as the RFSM, cannot model probabilities of occurrence; and are therefore unable to represent flood prevention measures (Scussolini et al., 2016) and distinguish defended and undefended floodplain zones. Much of the exposed population in the Netherlands, as well as other countries with flood protection, reside in the defended area of a

275 floodplain. This does not eliminate their risk of flooding; just reduces the probability of it. The severity of a flood event when defences fail can be catastrophic, resulting in high velocity flows and rapid inundation with little to no warning.

The top 50 exposed countries calculated using the WorldPop and HRSL datasets are detailed in *Figures S13* and *S14*, respectively. We also compare continental and global flood exposure estimates from different sized rivers calculated using GHS-POP and WorldPop in *Table 1*. It's not possible to compare these global results with HRSL calculated exposure,

as it does not yet have global coverage. Global exposure calculated using the WorldPop layer is 2.026 Billion, roughly 83 Million larger than the global figure calculated using GHS-POP. Differences in exposure between the two datasets are largest in Africa and Asia and Oceania. We explore the implications of using different population datasets for flood exposure calculations in greater detail in *Section 3.3* of this paper

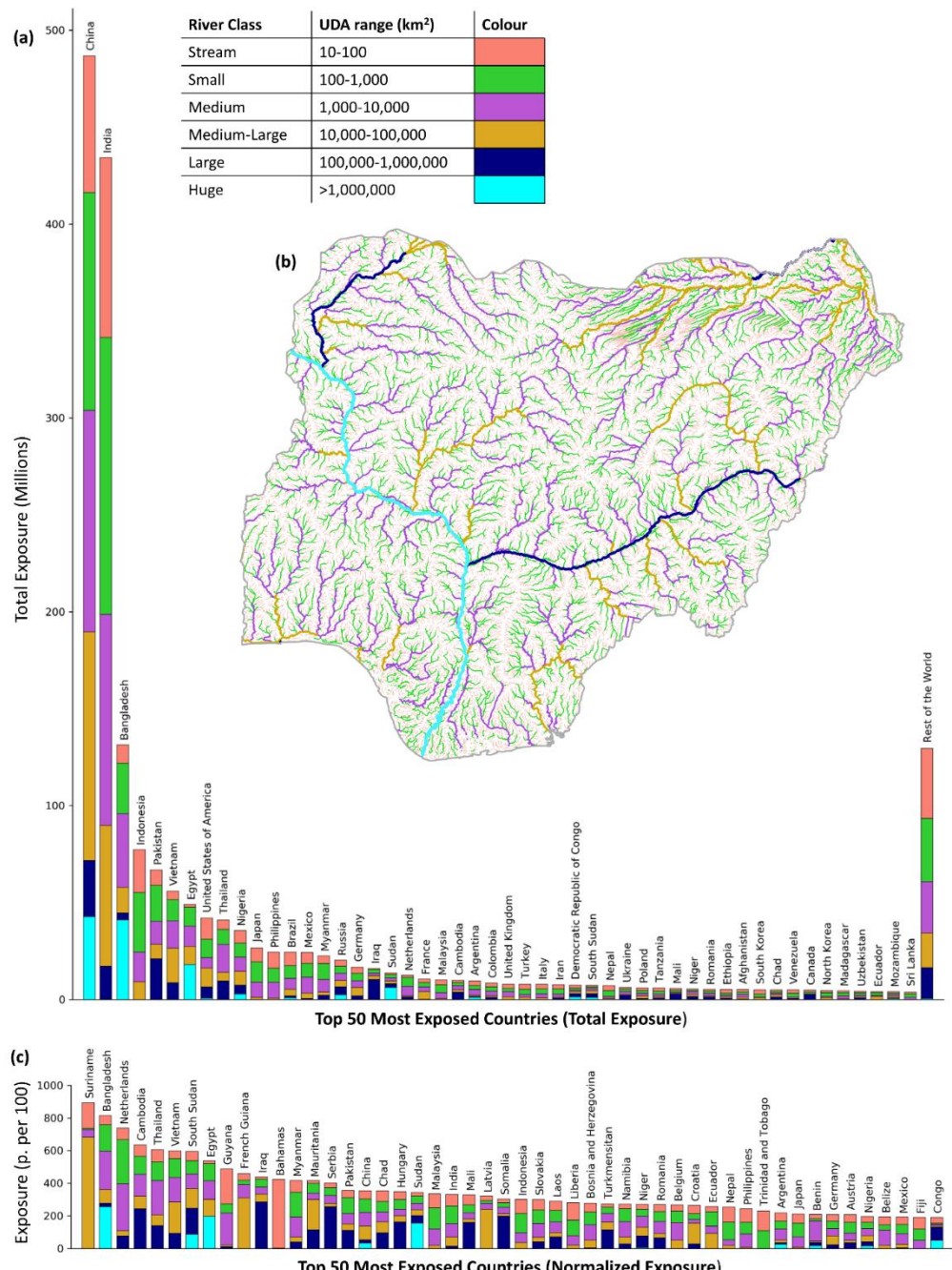

**Figure 3.** Flood exposure calculated with the Global Human Settlement Population (GHS-POP) layer (a) Top 50 most exposed countries in terms of total flood exposure. (b) The river size classifications visualized in Nigeria. (c) Top 50 most exposed countries in terms of normalized flood exposure (normalized to country's total population).

**Table 1.** Comparison of continental and global flood exposure estimates from different sized rivers calculated with Global Human Settlement Population (GHS-POP) layer and WorldPop. Exposure is in millions of people.

| River Class | Africa | | Americas | | Asia and Oceania | | Europe | | **Global** | |
|---|---|---|---|---|---|---|---|---|---|---|
| | GHS-POP | WorldPop | GHS-POP | WorldPop | GHS-POP | WorldPop | GHS-Pop | WorldPop | GHS-POP | WorldPop |
| Stream | 33.2 | 42.1 | 38.88 | 38.45 | 260.53 | 274.69 | 20.65 | 20.07 | 353.26 | 375.31 |
| Small | 41.03 | 48.43 | 36.72 | 36.13 | 409.21 | 415.31 | 26.63 | 26.01 | 513.59 | 525.88 |
| Medium | 39.41 | 43.45 | 29.84 | 30.28 | 363.67 | 384.77 | 26.84 | 26.72 | 459.76 | 485.22 |
| Medium-Large | 34.23 | 35.91 | 20.94 | 20.65 | 260.44 | 268.13 | 18.64 | 18.6 | 334.25 | 343.29 |
| Large | 25.36 | 21.8 | 11.9 | 11.9 | 114.14 | 126.46 | 11.4 | 11.5 | 162.8 | 171.66 |
| Huge | 30.45 | 30.24 | 2.65 | 2.74 | 86.41 | 92.09 | 0 | 0 | 119.51 | 125.07 |
| **Total** | 203.68 | 221.93 | 140.93 | 140.15 | 1,494.4 | 1,561.45 | 104.16 | 102.9 | 1.943.17 | 2,026.43 |

## 3.2 Exposure Change from 1975 – 2015

An advantage of both the GHS-POP and WorldPop datasets is their population estimates across different time scales, allowing for exposure analysis over time. WorldPop has annual population maps from 2000-2020 and GHS-POP has population estimates across four epochs: 1975, 1990, 2000, 2015. Here, using GHS-POP's multitemporal population layers, we calculate exposure change over a period of 40 years. Normalized flood exposure estimates were calculated for the years 1975, 1990, 2000, and 2015. These results are tabulated in *Table S10*. Population change is calculated by taking the difference between the normalized exposure estimates for the years considered. Globally, total flood exposure grew between 1975 and 2015 from 257 people per 1000 to 265 people per 1000. Interestingly, in both Tropical and Arid climates total flood exposure over this 40-year period grew by 11 people per 1000; but in Temperate and Continental climates total flood exposure decreased by 4 and 10 people per 1000, respectively. Developing countries are largely located in tropical and arid climates, conversely, developed economies are prevalent in temperate and continental climates. These findings correspond with previous work done by Jongman et al. (2012), which found developing countries had the largest increases in exposure relative to population growth in the period 1970-2010. At the continental level, normalized flood exposure saw the largest increase in Asia, growing by 15 people per 1000 from 1975-2015. It also grew in South America by 5 people per 1000. In Europe, changes in normalized exposure over this period were negligible; while in North America, Africa, and Oceania normalized exposure decreased by 3, 5, and 2 people per 1000, respectively. Comparing these results with a related study by Ceola et al. (2014), which used satellite night-time light intensity to explore changes in river flood exposure from 1992-2012, we find similar trends in North America, South America, Europe, and Asia. Exposure over the period 1975-2015 increased for streams, medium-large, large and huge rivers. There were slight reductions in exposure for small and medium sized rivers.

Exposure changes at the national level are depicted in *Figure 4*. The highest increase in overall flood exposure was seen in Nepal and French Guinea. In both countries, the proportion of exposed population grew by 200 people per 1000 in the period 1975-2015. In French Guinea, this sudden increase is largely due to the population growth of Saint-Laurent-du-Maroni, a town situated on the banks of the Maroni river. From 1975-2015 the town's population grew 1800% compared with the national population growth of 360%. In Nepal, one of the top 10 fastest urbanizing countries in the world (Bakrania, 2015), the flood exposure growth is a result of this fast urbanisation in cities such as Kathmandu, which is dissected by eight different rivers. An exposure decrease of 172 people per 1000 was seen in South Sudan. This is due to the growth of urban areas outside the Sudd swamp in cities such as Juba, Yei, Yambio, Nzara, and Wao. South Sudan has been hit by devastating floods in the past year, which displaced over 800,000 people (OCHA, 2020). Had relative population exposure in South Sudan grown, rather than shrunk, the recent flooding could have been even worse.

**Figure 4.** Country level river flood exposure (population normalized) change from 1975-2015 calculated using the Global Human Settlement Population (GHS-POP) layer. River size expressed in Upstream Drainage Area (UDA).

### 3.3 Exposure Estimates from Different Population Datasets

Exposure differences arising from the use of different population layers were calculated for the 168 countries where all three population datasets are available (*Figure 5)* (see *Table S11* for a list of the missing countries). In the countries examined, normalized exposure (with respect to the country's total population) calculated with WorldPop data was the highest (270 exposed per 1000), followed by GHS-POP (256 exposed per 1000), HRSL exposure was the lowest (235 exposed per 1000). These findings correlate with a previous study by Smith et al. (2019) which found WorldPop data
overestimated flood exposure compared to HRSL data in each of the 18 developing countries examined.

        Differences in calculated exposure across the river sizes are shown in *Figure 5b*. Exposure differences were most pronounced for smaller rivers (streams, small, and medium rivers), while there was almost no exposure difference for the largest river class (huge).  The overall trend across all river sizes consistently shows that WorldPop estimated the highest exposure, followed by GHS-POP, and then HRSL with the lowest.

The  population mapping approaches of the three population layers can go some way towards explaining the differences in calculated exposure, these corresponding outputs are visualized in *Figure 6*, where we qualitatively compare the population distribution of the three outputs with respect to the settlement distribution, manually identified from high-resolution satellite imagery, along the Likuala-aux-Herbes river in the Republic of Congo. WorldPop's population distribution algorithm dasymetrically redistributes the whole population across the grid, also in areas where no settlements
have been identified. This is done under the assumption that not all 'built up' areas will be picked up in the satellite imagery (TReNDS, 2020). When intersected with a flood extent, such a modelling approach can lead to mis-estimation of flood exposure in rural areas with respect to the other two population datasets. In the area examined in *Figure 6*, WorldPop estimates 1,167 people exposed, compared with 17,581 and 13,789 people exposed estimated by HRSL and GHS-POP, respectively. This is despite WorldPop exposure covering over 93% of the area examined, which far exceeds GHS-POP's
5% exposed area and HRSL's 1% exposed area. WorldPop's approach to rural population distribution can lead to underestimation of exposure in small rural settlements (such as in *Figure 6*) or overestimation of exposure across large expansive areas of flooding, as will be explored later in this section. Conversely, the approach implemented by both GHS-POP and HRSL (which spread census data only over identified 'built up' areas) is more sensitive to omission and commission errors arising from the classification of settlements (Palacios-Lopez et al., 2019). For example, undetected
settlements outside the flood extent would result in artificially higher flood exposure estimates as the underlying census data is only spread across the identified settlements (an incorrectly greater proportion of which are now identified as within the flood extent). Similarly, commission errors (false positives) are common in sandy or rocky landscapes and often occur in coastal areas or along riverbanks. Commission and omission errors can lead to either artificial increases or decreases in flood exposure estimates, depending on the location of these errors with respect to the flood extent.

The resolution of the population layers should also be considered. GHS-POP's fairly coarse (9 arc second) resolution means that in some areas where the potential for flooding (or not) falls within the resolution of a 9 arc second grid

cell, the settlement's avoidance (or not) of the flood risk cannot be accurately represented. This effect can be reduced by upsampling and proportionally reallocating the population to a grid that matches the resolution of the flooded data, as we have done in this study. Similarly, the spatial resolution of the underlying satellite imagery should be considered. Both GHS-POP and WorldPop identify settlements using Landsat imagery at 30 m resolution, while HRSL identifies settlements using DigitalGlobe imagery at 0.5 m resolution. Previous work by Tiecke (2017) showed that HRSL was able to identify buildings missed by GHS-POP, highlighting the importance of high-resolution imagery for comprehensive building classification.

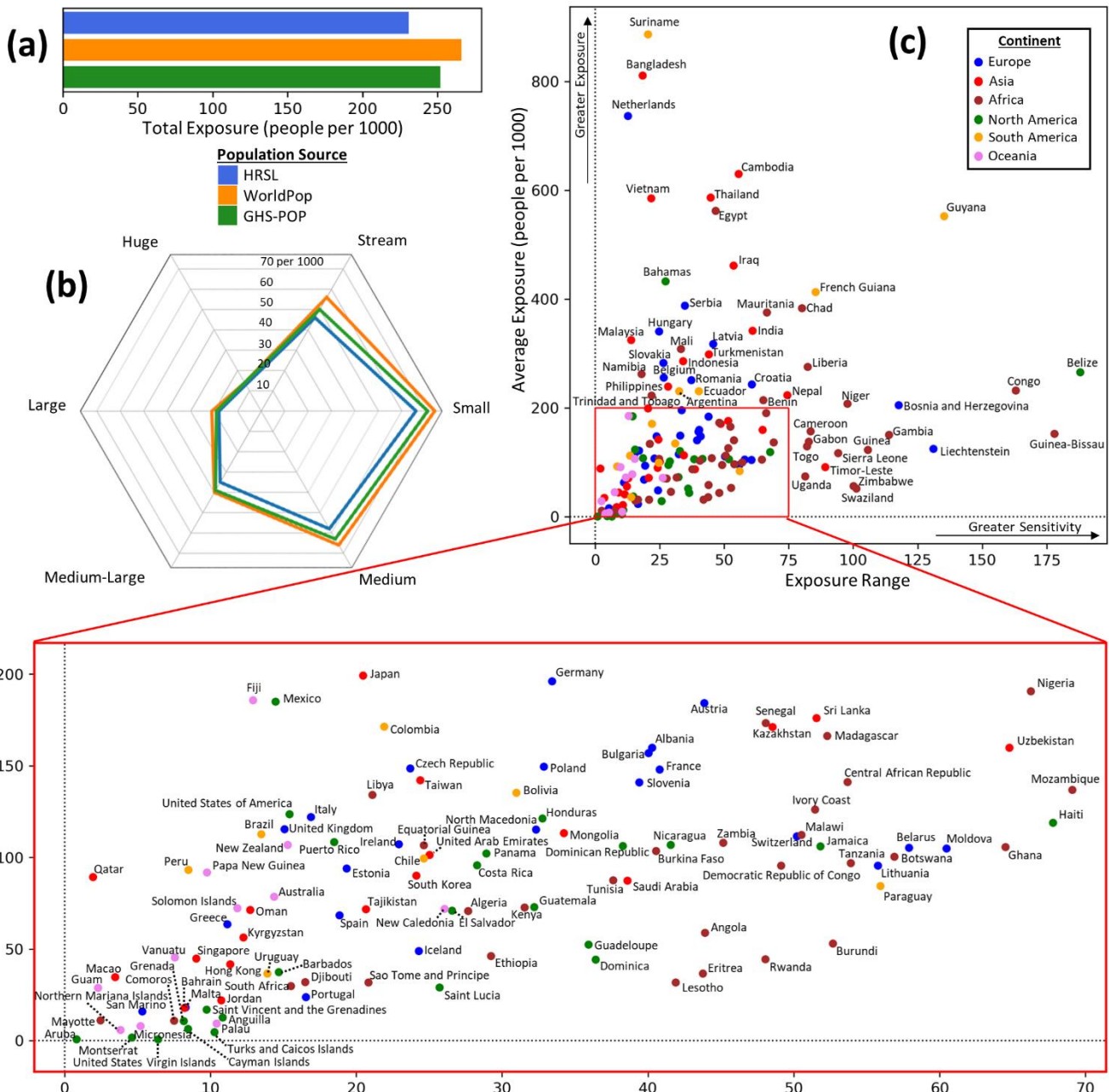

**Figure 5.** Flood exposure comparison in 168 countries using the High Resolution Settlement Layer (HRSL), WorldPop, and Global Human Settlement Population (GHS-POP) layer. (a) Comparison of the total normalized flood exposure between the three population datasets in all available countries. (b) How the calculated exposure figures differ per river size classifications. (c) Country level statistics for average normalized exposure (calculated as the mean of the three national exposure estimates) and the sensitivity of the exposure calculation to the choice of population dataset (measured as the absolute range of the three national exposure estimates). The higher up the Y axis and X axis, the greater the average exposure and sensitivity to the choice of population dataset, respectively.

The use of different population datasets had a negligible effect on exposure estimates for the huge river class. Large settlements tend to form around rivers of this size, and on coastlines where rivers of this size drain. Large urban areas are easily identifiable from remote sensing data, which means the population distribution (and resulting exposure estimates) for these urban centres show less variation between the datasets. Conversely, non-urban flood exposure estimates to smaller rivers show greater sensitvity to the choice of population layer. This is because the approach to non-urban population

mapping between the three datasets differ. WorldPop, as mentioned previously, distributes administrative level census data across all 3 arcsecond pixels in order to mitigate the impacts of potential omission and commission errors in the settlement data. This approach leads to some overestimation in rural populations (Smith et al., 2019, Wardrop et al., 2018). GHS-POP, which distributes census data over Landsat identified settlements (and in non built-up areas distributes population at the census unit by areal weighting), tends to underestimate rural populations. (Liu et al., 2020, Leyk et al., 2019). HRSL's use of

ultra-high resolution sattelite imagery has been shown in previous studies to accurately identify rural settlements (Tiecke, 2017, Smith et al., 2019). However, the method of proportional allocation used to distribute the census data is relatively crude. Uncertainties in the underlying census data should also be considered, as the quality and detail of the data, as well as the frequency at which it is collected, varies significantly at the national level (Leyk et al., 2019). The three population datasets compared in this study share the same input census data (GPWv4) and therefore any associated census uncertainties

are a common feature shared across the three datasets.

Calculating the general trends of exposure between the population layers is useful for making broad conclusions about the suitability of a population layer. Understanding the variations of the data at the country level leads to more actionable information about the appropriate use of different population layers. We calculate both the severity of flooding in each country (as the mean of the normalized national flood exposure estimates calculated with the three population datasets)

and the disagreement between the population exposure estimates in each country (as the absolute range of the three normalized national flood exposure estimates). The disagreement between the population layer exposure estimates for each country varies significantly (*Figure 5c*). In the three countries with the highest exposure disagreement (Belize, The Republic of Congo, and Guinea-Bissau) WorldPop estimates of exposure are far greater than either HRSL or GHS-POP estimates. In Belize, a country with large areas of inundated wetlands, WorldPop estimates 135,000 people exposed, while GHS-POP and

HRSL estimate 70,000 and 80,000 exposed, respectively. In the Republic of Congo, a country with large areas of floodplain, WorldPop estimates 1.3 Million people exposed and GHS-POP and HRSL estimate 810,000 and 780,000 exposed respectively. WorldPop's method of distributing the population over a large area results in significant overestimation compared with HRSL or GHS-POP in these rural inundated areas. This can be seen in greater detail in *Figure 7* for Guinea-Bissau. In Guinea-Bissau, GHS-POP and HRSL (which estimate exposures of 180,000 and 160,000 respectively) identify

settlements largely situated outside the floodplains ('dry' cells in blue). Comparatively, WorldPop's modeling approach and assumptions leads to far more 'wet' population cells and an estimate of exposure (480,000) more than double that of the other two population layers. The exposure disagreement in these three countries is compounded by the relatively large areas of inundation in each country. The percentage inundated area is 25%, 30%, and 26% for Guinea-Bissau, Belize, and The

Republic of Congo, respectively. In comparison, the percentage of populated area defined by the population layers is less than 5% for GHS-POP and HRSL, but more than 95% for WorldPop in each of the three countries. As exposure in this study is defined as the intersection of the flooded area and the populated area, it is understandable that WorldPop's exposure estimates are more sensitive to the area of inundation. This is evident when examining a country with high exposure disagreement but with a comparatively smaller area of inundation. In Bosnia and Herzegovina *(Figure 7)*, the percentage of flooded area is just 9% and the GHS-POP layer estimates far greater exposure (1 Million) than either WorldPop (680,000) or HRSL (610,000). Here, where much of the exposure occurs near the banks of the rivers, the coarse spatial resolution of GHS-POP is less able to precisely locate settlements situated just outside the floodplain. As a result, more populated cells are flagged 'at risk' compared to the higher resolution HRSL layer.

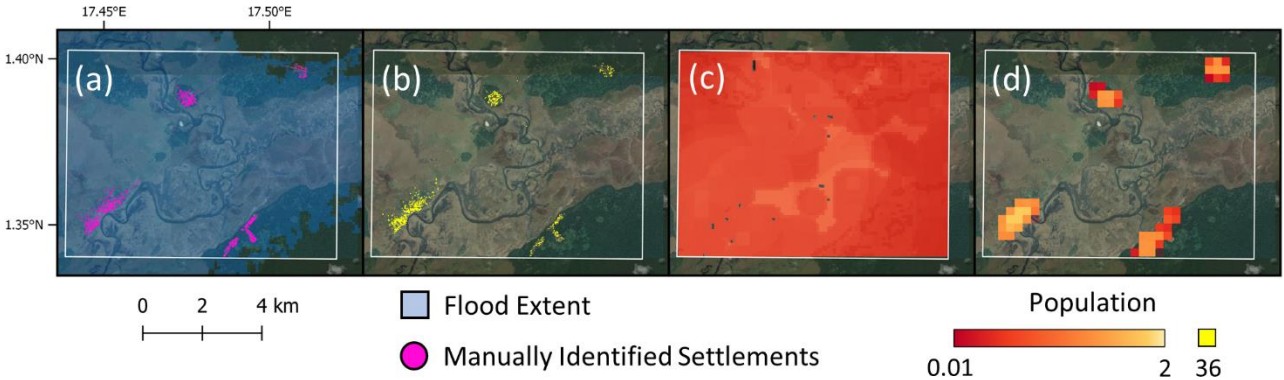

**Figure 6.** Qualitative comparison of settlement distributions on the Likouala-aux-Herbes river in the Republic of Congo. The white square in each panel is the pre-defined bounding box for which population totals are calculated. Population pixels in panels (b)-(d) range from low populated pixels (red) to high populated pixels (yellow) (a) River Flood Susceptibility Map (RFSM) flood extent (blue pixels) along with manually identified settlements (pink circles) from high resolution Google Earth satellite imagery. (b) High Resolution Settlement Layer (HRSL) population distribution. 17,581 people exposed. (c) WorldPop population distribution (resampled to 1 arc second for comparison). 1,167 people exposed. (d) Global Human Settlement Population (GHS-POP) layer population distribution (resampled to 1 arc second for comparison). 13,789 people exposed. Map Data: © Google, Maxar Technologies 2021

These results have shown that the use of different population layers can lead to vastly different flood exposure estimates because of inherent differences in their spatial resolutions, methods used, and assumptions made to produce them. Our comparative analysis has identified in which countries exposure calculations are sensitive to the choice of population layer and shed light on some of the reasons for exposure disagreement. However, there is a limit to the conclusions that can be drawn from comparative analyses alone, and there is an urgent gap for more studies which validate the accuracy of these population layers using ground-truthed data.

It would be imprudent to definitively recommend one population dataset for use in flood exposure studies without extensive comparative global validation. However, previous studies have shown that HRSL performs better than existing population datasets at mapping reference building footprints, especially in rural areas (Tiecke, 2017, Smith et al., 2019). Our results also point to some of the benefits of using HRSL. Its settlement identification method for population distribution avoids exposure overprediction common in other population data and its high resolution can better capture the accurate

location of settlements. Despite this, HRSL shouldn't be considered a catchall dataset for flood exposure. Its high resolution may limit its use in certain situations due to computational restraints. Similarly, in studies of flood risk over time population data with multiple temporal epochs, such as GHS-POP or WorldPop, are better suited. The results we present in this section, and *Figure 5*, are intended to inform users of these population datasets about their appropriate use. In countries with high exposure disagreement, the choice of population dataset for flood exposure should be carefully considered, and further accuracy assessments of the population layers are recommended.

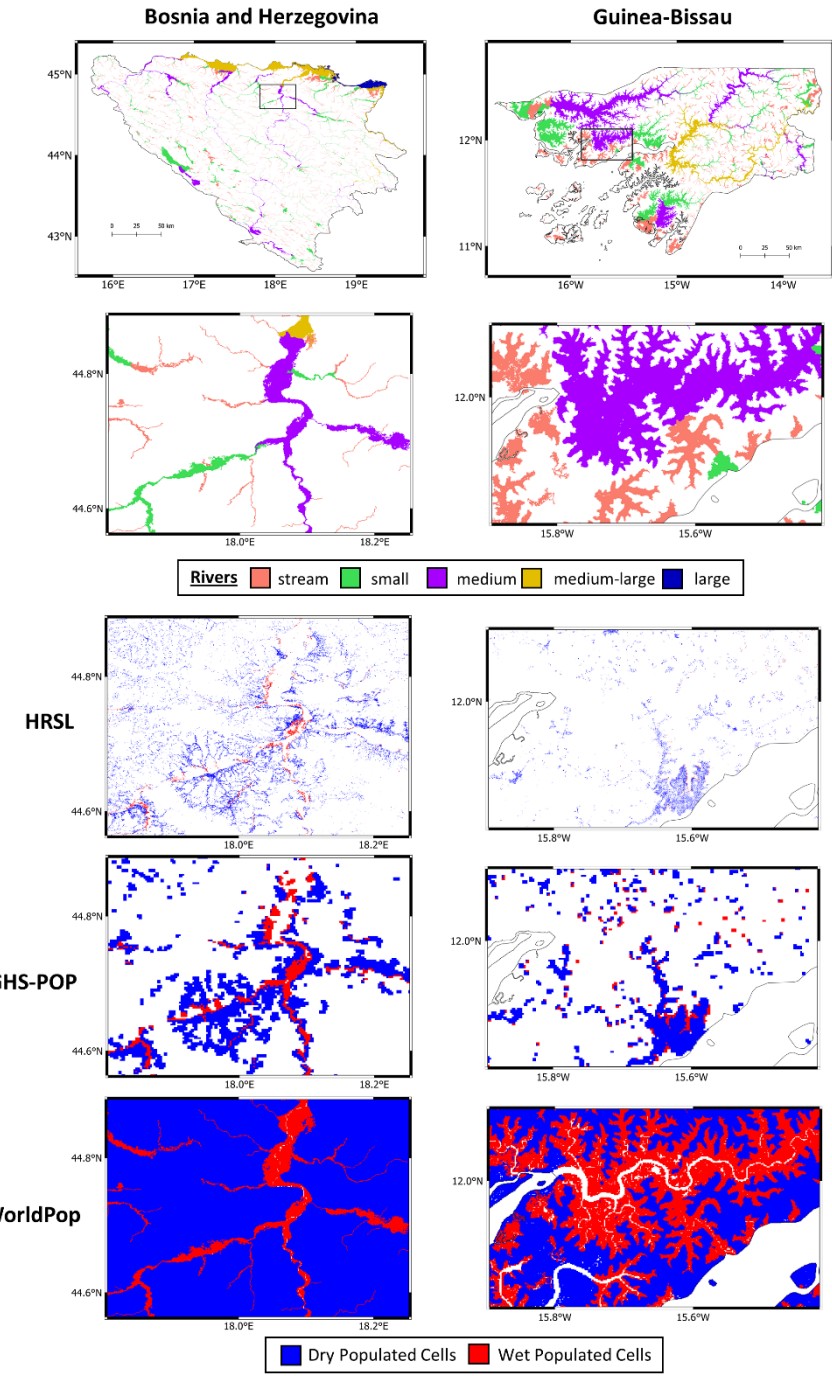

**Figure 7.** Comparison of population datasets and their intersection with the flood extent in Bosnia and Herzegovina and Guinea-Bissau. The top two windows show the River Flood Susceptibility Map (RFSM) split into the different river size categories for the whole country (top panel) and for a smaller, more detailed area of both countries (second panel from top). The remaining windows show the three different population maps and their intersection with the flood map in the detailed areas of both countries. Blue cells indicate the population cells are dry (not exposed to flooding) and red cells indicate the population cells are wet (exposed to flooding).

**3.4 Relevance to Global Flood Models**

The minimum size of river represented in Global Flood risk Models (GFMs) varies (see *Table 2*), with minimum river size thresholds ranging between 50-5000 km$^2$ UDA, three orders of magnitude. River network size can be limited by the granularity of input data such as rainfall (Dottori et al., 2016c), or by the computational demand of modelling floods at the global scale.

**Table 2.** Global flood model river representation

| Minimum River Size (upstream drainage area) | Global Flood Risk Model | River Sizes Modelled (P = Partial) |
|---|---|---|
| 50 km$^2$ | Fathom (Sampson et al., 2015) | Stream (P), Small, Medium, Medium-Large, Large, Huge |
| 500 km$^2$ | ECMWF (Pappenberger et al., 2012) and U-Tokyo (Yamazaki et al., 2011) | Small (P), Medium, Medium-Large, Large, Huge |
| 1000 km$^2$ | CIMA-UNEP (Rudari et al., 2015) | Medium, Medium-Large, Large, Huge |
| 5000 km$^2$ | JRC (Dottori et al., 2016c) | Medium (P), Medium-Large, Large, Huge |


Differences in river network size between GFMs undoubtedly lead to differences in global flood exposure estimates. These differences can be even more pronounced at the national level, where GFMs have been used to inform disaster risk management (Ward et al., 2015). Flood exposure was calculated for the different GFM river thresholds using the GHS-POP layer. Globally, we found that exposure estimates between the river threshold which results in the largest river

network (>50 km$^2$ UDA), and the river threshold which results in the smallest river network (>5000 km$^2$ UDA), differ by over a factor of 2. If the size of the river network was further increased by reducing the river threshold to 10 km$^2$ UDA (below current GFM representation), the exposed population captured increases by 13%.

At the national level, in countries such as Suriname, The Republic of Congo, and Egypt, the greatest proportion of flood risk is posed by rivers with a UDA of 5000 km$^2$ or greater. In these countries, GFMs could be used interchangeably.

Understanding what size rivers pose a significant flood risk is key to accurately representing national flood risk. In Benin, for example, the estimated flood exposure when a 5000 km$^2$ UDA threshold is applied is 0.49 million people. When the threshold is reduced to 1000 km$^2$ UDA, the estimated exposure increases to 1.8 million people. Some countries do not have large rivers flowing through them, and the flood risk will result entirely from smaller rivers. Often these are island nations, such as in Jamaica or Trinidad and Tobago, where all flood risk is from rivers smaller than UDA 1000 km$^2$. However, in

Andorra for example, a landlocked country, to capture any flood exposure, a 50 km$^2$ UDA threshold is needed.

To aid national level flood risk practitioners in their choice of GFM, we calculated the minimum river threshold required to capture a given percentage of the largest river network's (>50 km$^2$ UDA) national exposure. Exposure percentages ranging from 10-90% were calculated for each of the three population datasets used in this study and mapped for each nation, globally. All 27 maps are included in *Figures S15-S17*. *Figure 8*, which shows the minimum river threshold

required to capture at least 50% of possible GHS-POP exposure, illustrates these results. The map shows that while in some countries GFMs could be used interchangeably, in others, the size of the river network could significantly impact national flood exposure estimates.

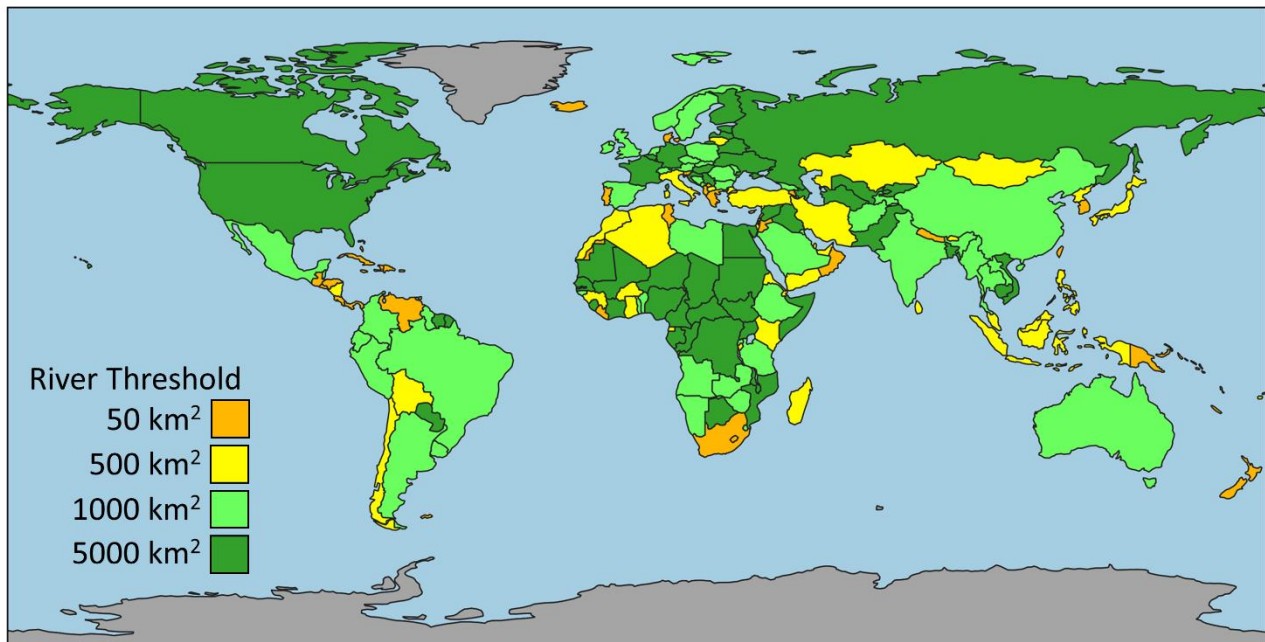

**Figure 8.** In which countries is the choice of river threshold important? The map shows the global flood model (GFM) river upstream drainage area (UDA) threshold required to capture over half a country's total flood exposure. In dark green countries the choice of threshold is less important than in orange countries. Grey areas are no-data regions. The map was calculated using the Global Human Settlement Population (GHS-POP) layer. See *Figures S15-S17* for maps calculated with the other two global population layers and for different percentages of total national exposed population.

It is difficult to exhaustively compare global flood exposure estimates from previous GFM studies as often exposure is expressed differently (e.g. expected annual exposure (EAE) vs. exposure to a return period flood) and sometimes global exposure is not reported at all. In the comparable studies, there is significant variation in global flood exposure estimates. In Ward et al. (2013) global EAE was calculated at 169 Million. This figure is almost triple the 58 Million calculated by Dottori et al. (2018) and the 54 Million calculated by Alfieri et al. (2017). In studies reporting exposure to a 100-year flood, Hirabayashi et al. (2013) estimate 847 Million people exposed and Jongman et al. (2012) estimate 805 Million exposed.

The need for independent model comparison studies was met by Trigg et al. (2016b) and Aerts et al. (2020) who compared GFM output in Africa and China respectively. These studies compared the output of multiple GFMs, finding large disagreement between the modelled flood extents. Both studies also found large variations in calculated exposure. However, differences in exposure calculated by the GFMs were found to be influenced just as much by different model forcing and resolution as by differences in river network size. Uncertainty in GFMs needs to be explored across the model cascade to identify where the models need to improve. Studies such as Zhou et al. (2020), which explores uncertainty in model forcing; and this study, which explores uncertainties in river network size, are important steps in directing future model development.

Granularity of input data is the main obstacle to increasing river network size in GFMs. The terrain data in all these models, which strongly influences their performance, is derived from the Shuttle Radar and Topography Mission (SRTM), a mission over two decades old (Farr et al., 2007). New, 1 arc second resolution (~30 m at the equator) global DEMs have recently been released by both the National and Aeronautics and Space Administration (NASA) and the European Space Agency (ESA). The ESA DEM is particularly important as its elevation is based on newer satellite data from TanDEM-X. A new method for deriving an elevation map from satellite images has also been developed by Google, capable of generating DEMs at 1m resolution (Nevo, 2019). Whether it is terrain or climatology data, new and improved methods are constantly being developed and better datasets are being released. There is scope in the near future for increasing river network size in GFMs. This comes at a computational cost, however; whether it's the use of a higher resolution DEM or the exponential increase in number of rivers to model when the threshold river size is reduced. Understanding where the representation of smaller rivers is needed most, namely in areas of high exposure, would streamline the future development of GFMs, targeting improvements in areas where flood risk is highest.

## 4 Conclusions

This study has presented the first global picture of flood exposure categorised by different sized rivers. We introduced a simple geomorphological approach to delineating a river's flood susceptibility, which is suitable for global scale 'first look' studies such as this and importantly, allows an assessment of river network size independent of global flood model structural and computational limitations. We find that over 75% of the global flood exposure is in Asia, with China and India making up a significant proportion of this total. Streams (UDA 10-100 $km^2$) and small rivers (UDA 100-1000 $km^2$) are responsible for over half of India's flood risk. At the global scale, these rivers contribute to 45% of total flood exposure, emphasizing the importance of the incorporation of these smaller rivers into global flood risk studies. We find that large increases and decreases in flood exposure over the last forty years are a result of urbanisation, either inside the flood risk zone or outside of it. The effect that the choice of population dataset had on exposure calculations differed between countries. Globally, this effect was most pronounced on smaller rivers, suggesting future studies that incorporate these smaller rivers should be careful in their choice of population data. Global flood models, the current tools for examining global flood risk, differ significantly in the size of their river networks. We found that the global flood exposure estimates differed by greater than a factor of 2 when calculated using the GFM river threshold which results in the largest river network (UDA >50 $km^2$) compared to the river threshold which results in the smallest river network (UDA >5000 $km^2$). These differences were often more pronounced at the national level.

The results of this study are intended to inform both the developers and users of global river flood models. Consideration of river network size, and how this relates to exposure, is imperative to having a comprehensive picture of flood risk. Increasing the size of the river network comes with both data and computational restraints. Doubling the resolution of the models (from 1km to 90 m to 30 m) requires an order of magnitude increase in computing power. Finer

resolution grids are imperative for representing small streams accurately. This has big implications for models currently operating at coarse resolution. Modelling smaller rivers requires not only detailed high-resolution data, but also efficient modelling structures capable of running at higher resolutions. Understanding where the representation of small rivers is needed most (areas of high exposure) can focus future model development. Similarly, accurate flood exposure estimates necessitate accurate population data. We have shown that the choice of population data used in exposure calculations can have an enormous impact on flood exposure estimates and we have identified in which countries this disagreement is most extreme and have identified some of the reasons for this. Flood risk practitioners should use these results as guidance about which population layer is best suited for their locality and use. There is need for further research in this area, incorporating more population data, as these layers play such an integral role in flood exposure calculations. In addition to more comparative analyses, there is also an urgent need for this population data to be validated at the global scale with actual data collected on the ground. Only then can definitive conclusions be drawn about the appropriate use of different population datasets. The selection of GFMs available to the end user is large and increasing. However, differences in the size of river networks between the models can have a significant impact on flood exposure estimates. While available GFMs could be used interchangeably in some countries, in others, discrepancies in river network size would lead to vastly different national flood exposure estimates. The results of this study should help to inform GFM users about the appropriate choice of GFM for their country of interest.

**Author Contributions**

MVB and MAT conceived of the study. MB designed and carried out the analysis. MAT PAS CCS AMS supervised the project. MVB drafted the manuscript. All authors contributed towards the discussion and editing of the manuscript.

**Competing Interests**

The authors declare that they have no conflict of interest.

**Acknowledgements**

MVB was funded by the UK National Environmental Research Council Grant NE/R008949/1 and iCASE funding from Fathom Global. This work was also supported by the Water Security and Sustainable Development Hub funded by the UK Research and Innovation's Global Challenges Research Fund (GCRF) [grant number: ES/S008179/1]. This work was undertaken on ARC3, part of the High-Performance Computing facilities at the University of Leeds, UK. The authors would like to thank the members of the Global Flood Partnership, who have helped to shape this research through discussion and feedback at numerous GFP workshops.

## Data Availability

All the data used in this study is freely available to download. The River Flood Susceptibility Maps are available from the University of Leeds at https://doi.org/10.5518/947. Facebook's High Resolution Settlement Layer can be downloaded by following the instructions in this link https://data.humdata.org/organization/facebook?q=density . The GHS-POP data can be downloaded here http://doi.org/10.2905/0C6B9751-A71F-4062-830B-43C9F432370F. The WorldPop data can be downloaded here https://dx.doi.org/10.5258/SOTON/WP00645.

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
