# Peer review of "Global Flood Exposure from Different Sized Rivers"

_Natural Hazards and Earth System Sciences, 2021_

## Author Response (AR1)

**Reviewer 1**

General comments:

The manuscript provides detailed and useful information about the importance of selecting a "good" and reliable population dataset to assess exposure to floods at global scale. It also present an alternative approach to improve flood susceptibility mapping, by means of a simple geomorphic variable. The paper is well written and enjoyable. Results and comments are significant for future applications. I believe the paper can be published after complying with minor issues.

***Author's Response:*** *We would like to thank Serena Ceola for her in-depth review of our manuscript and we appreciate her positive comments about our paper. We feel that her suggestions have significantly improved the manuscript and we outline the changes we have made below.*

Specific comments:

l. 9. I would suggest to cite RFSM here

***AR:*** *Agreed, we have spelled out the RFSM in the abstract (lines 9-10).*

ll. 32-34: the authors may refer to Ceola et al., 2014, GRL, https://doi.org/10.1002/2014GL061859, (where nighttime lights are used to assess human exposure to floods, including also temporal trends. It may be interesting to compare results (see exposure change from 1975-2015 and Fig. 4).

***AR:*** *Thank you for pointing us to this paper. We have compared the results of our paper with the Ceola et al. paper in the section "Exposure Change from 1975-2015" (lines 339-344).*

ll. 49-64: this part looks like a repetition of waht was written before. I would suggest to remove it or rephrase it.

***AR:*** *We agree there is repetition of points already addressed earlier in the introduction. We have removed the paragraph in question (lines 56-71) and added relevant information to the previous paragraphs in the introduction (lines 27-33 and lines 53-55).*

ll. 100-105: authors should check the paper written by Samela et al., 2015, AWR, https://doi.org/10.1016/j.advwatres.2017.01.007, where a geomorphic index (GFI) is introduced to define a flood susceptibility map. A thorough comparison should be performed, commented and included in the revised version of the mansucript.

***AR:*** *This is indeed an important paper relevant to our work that should be commented on and referenced in our manuscript. Thank you for pointing it out. We have compared our approach to the Samela et al. (2015) paper as well as another relevant paper referenced in Samela et al. (2015). We compare the approaches in lines 113-122 as well as lines 160-162.*

Figure 1: I would suggest to show an example with a 10 km2 threshold derived from analyzed data - currently this Figure assumes a different UDA that is confusing.

***AR:*** *This is a very good point. We have updated Figure 1 to use a 10 km2 threshold. To do this we needed to increase the size of the illustrative grid from 10x10 to 12x12. We also pointed the reader to Figure 7 in the caption, which shows actual RFSM output in Bosnia and Herzegovina and Guinea-Bissau.*

ll. 139-140: authors should better clarify why Hn is needed. Also, clearly separate calibration and validation by adding e.g. a chartflow or better rephrasing the text, listing in detail the areas used for calibration and validation respectively.

*AR: Thank you for pointing this out. We have updated the manuscript to include two new sub-sections that separate the calibration and the validation of the RFSM. In lines 159-162 and lines 188-191 we clarify why Hn is needed. We have also worked to clarify the text in both sections. We include more detail about the calibration and validation regions (lines 174 - 187 and lines 221-224). We have also added three columns to Table S1 which contain more information about each of the calibration basins.*

l. 144 and Figure 2: authots should provide here a list of the 19 reference flood maps and substitute Figure 2 with Figure S1.

*AR: Good point. We have used the map from Figure S1 in Figure 2 and have removed Figure S1 from the supplementary material. We have also listed the flood maps used for calibration in lines 174-187.*

l. 161: add a list of 6 GFMs

*AR: We have added these on lines 205-208*

l. 166: which kind of commonly used measure of fit scores did the authors use? Please list them here

*AR: We have spelled out and referenced the measure of fit scores we used in lines 189-190*

l. 208: HRLS and World Pop data: to which year do they refer?

*AR: Very good point. They refer to the years 2018 and 2015, respectively. We have added this in the manuscript on line 266*

l. 211: add "from GHS-POP" in the title

*AR: This is a good point. We have updated the title.*

l. 243: authors should cite the paper written by Scussolini et al., 2016, NHESS, https://doi.org/10.5194/nhess-16-1049-2016 on a global-scale flood protection database

*AR: Yes we agree. This paper has been cited on line 300 of the updated manuscript*

ll. 245-254: authors may consider to remove this part and simply refer to section 3.3

*AR: This is a good point. We have removed the paragraph in question (lines 311-320) and have pointed the reader to section 3.3. We've also included a short paragraph (lines 304-310) commenting on Table 1, which we added based on the comment below.*

ll. 248-254: authors should add a table to show a comparison between GHS-POP, World POP and HRSL and should cite Figures reported in the SI.

*AR: This is a very good point. We have added Table 1 which compares the continental level flood exposure results between GHS-POP and WorldPop. HRSL cannot be compared with the other two datasets at the continental level as it does not yet have coverage in all countries. Therefore, we point the reader to section 3.3 where we compare the three datasets in depth in the 168 countries where all three are available. We also point the reader to the additional figures in the supplementary on lines 304-305.*

Figure 3: I would suggest to avoid the use of acronyms in the figure caption. Also, I would suggest to start the caption as follows: "Flood exposure from ..."

*AR: We agree. The caption for Figure 3 has been updated. We have also updated the captions on all other figures to spell out any acronyms (Figure 1, Figure 3, Figure 4, Figure 5, Figure 6, Figure 7, Figure 8).*

ll. 268-271: as stated above, authors could compare their own results with temporal trends as in Ceola et al, 2014, GRL. Also, how change is computed? Is it simply a difference between 1975 and 2015?

*AR: We have compared our results with those of Ceola et al (2014) on lines 339-344. Exposure change is calculated as the difference in normalized exposure between 1975 and 2015. We have explained this on lines 332-333. We have also included in the supplementary material a table (Table S10) which reports the flood exposure results for all four years of the analysis 1975, 1990, 2000, and 2015.*

l. 286: the title "Variation in exposure" could be misleading - what about "Exposure estimates from different population datasets"?

*AR: This is a good point. We have changed the title for section 3.3.*

l. 287: what is the exact number of countries considered here? 168 or 169 (as written in Fig. 5 caption - please check this)

*AR: Thank you for pointing this out. It should read 168. We have updated the caption in Figure 5.*

ll. 297-299: I found this sentence unclear.

*AR: Agreed. We have rephrased this sentence (lines 373-376).*

Figure 5: dots are too small to be seen and distinguished. Authors should enlarge dot size in panel (C) and line size in panel (b). How are average exposure and exposure range computed? This information should be added to the main text. Is the exposure range a % difference? Is it normalized with respect to the country population?

*AR: This is a good point. We have updated Figure 5 to enlarge the dots in panel (c) and the lines in panel (b). Exposure in Figure 5 is normalized with respect to a country's population (clarified on line 364). We also clarify how average exposure and exposure range are calculated both in Figure 5's caption and on lines 430-433.*

Figure 6: what is the meaning of the white square in each panel? Blue pixles in panel (a): what do they represent? What is the amount of the total population in (b), (c) and (d) in the flooded area? Also, consider to explain colors in the caption.

*AR: Thank you for pointing this out. The white square is the bounding box for this analysis. We have added this clarification into the caption for Figure 6. We have also clarified in the caption what the colours mean in each of the panels. The number of exposed people in each panel is included in the figure caption and we have commented on these numbers in lines 383-387. We have removed lines 440-443 to avoid repeating ourselves.*

Figure 7: it would be helpful to zoom over the squared area and show more in detail the RFSM. Also, even though I appreciate the effort to differentiate population per cell, I would suggest to simply distinguish between wet and dry cells.

*AR: Good point. We have updated Figure 7 to include an additional row (second from top) showing a more detailed RFSM flood extent. We have also updated the bottom three rows of maps to show only wet/dry cells rather than population per cell.*

Figure 8: rephrase the caption

*AR: Agreed. We have rephrased the caption for Figure 8.*

Technical corrections:

l. 161: remove "the" before African

*AR: Thank you. We have updated line 205.*

l. 217: remove "are" before susceptible

*AR: Thank you. We have updated line 275.*

l. 262: a number is missing in "200-2020"

*AR: Thank you. We have updated line 329.*

l. 279: remove an extra dot

*AR: Thank you. We have updated line 353.*

l. 317: (Tiecke, 2007) should read Tiecke (2007)

*AR: Thank you. We have updated line 401.*

l. 392-393: remove "the" before population and write "cells" instead of "cell". Maybe write 3 arc sec instead of 30 m (for consistency)?

*AR: Thank you. We have updated the caption for Figure 7. We have removed the line about resampling because we are no longer looking at population per cell (just wet/dry cells) so resampling here is irrelevant.*

l. 445: its?

*AR: Thank you. We have updated line 550.*

**Reviewer 2**

The manuscript describes a scientifically-sound and relatively-easy-to-implement geomorphological approach to assess global flood exposure (over time) to different sized rivers. Global and national flood exposures are estimated using three different gridded population distribution products - differing in terms of their spatial resolution, the underlying assumptions made, and the methodology used to produce them. Results are compared and used to inform on (1) how the use of different river network sizes impacts both global and national flood exposure estimates, and (2) the appropriate application of the considered population distribution datasets.

I am very supportive of the Author's effort and would like to highlight that more comparative studies like this one should be conducted, especially at the intersection between population mapping and natural hazards and risks. The manuscript is timely, appropriate for the journal, and potentially of interest for its readers. It is well written, articulated and presented, and offer an original

contribution in the field of flood exposure, as well as valuable insights into the advantages and challenges of using gridded population datasets to assess exposure to hazards.

In my opinion, the manuscript should be published after minor revisions aimed at addressing the detailed comments below. I have really enjoyed reading the manuscript and want to congratulate the Authors for their work.

***Author's Response:*** *We are very thankful to the anonymous reviewer for their positive review and for their comments and suggestions which we feel have led to an improved manuscript. We have outlined all the changes we have made below.*

60: "Recent advances in population data, providing more detail and employing new modelling techniques" – I would suggest to rephrase this as "Recent advances in population mapping, providing a better and more detailed representation of the spatial distribution of the population, have been shown to drastically reduce flood exposure estimates in developing countries (Smith et al., 2019).

***AR:*** *Thank you for pointing this out. In light of the first reviewer's comment about repeating ourselves in the introduction we have removed the paragraph containing this sentence entirely (lines 56-71).*

84: "https://dataforgood.fb.com/docs/high-resolution-population-density-maps-85demographic-estimates-documentation/" – The provide link is not working.

***AR:*** *It seems the link has gone down since we submitted. Thank you for catching this. We have updated the link for the Facebook HRSL data to (https://data.humdata.org/organization/facebook?q=density). The link has been updated on line 91, line 239, and line 608.*

298: "these methods" – Should be "the corresponding outputs".

***AR:*** *Good point. We have updated line 374.*

298: "the settlement distribution of the three population datasets along the Likuala-aux-Herbes river in the Republic of Congo." – I would suggest to rephrase as follow: "the population distribution of the three outputs with respect to the settlement distribution, manually identified from high-resolution satellite imagery , along the Likuala-aux-Herbes river in the Republic of Congo"

***AR:*** *Thanks for suggesting this. We think this really helps clarify the text and have updated lines 375-376 accordingly.*

300: "algorithm spreads some residual population across the grid in areas where no settlements have been identified" – please rephrase as follow: "algorithm dasymetrically redistribute the whole population across the grid, also in areas where no settlements have been identified"

***AR:*** *Thank you for pointing this out. We have updated lines 378-379.*

302: "this residual population spread" – please rephrase as follow: "such modeling approach"

***AR:*** *Thank you. We have rephrased this on line 381.*

336: "there is still significant uncertainty in the underlying census data" – This represent a common feature shared by all three population datasets considered in this study (which are all using exactly the same input census data).

*AR: This is a good point. We have removed the sentence and included a short discussion on the uncertainty of the underlying census data on lines 423-427.*

354: "WorldPop's residual population spread leads" – please rephrase as follow: WorldPop's modeling approach and assumptions leads"

*AR: Thank you. We have rephrased line 446.*

Figure 6: "(b) HRSL settlement distribution. (c) WorldPop settlement distribution (resampled to 1 arc second for comparison). (d) GHS-POP settlement distribution (resampled to 1 arc second for comparions)." – Should be ""(b) HRSL population distribution. (c) WorldPop population distribution (resampled to 1 arc second for comparison). (d) GHS-POP population distribution (resampled to 1 arc second for comparions)."

*AR: Thank you for pointing this out. We have updated Figure 6 accordingly.*